# Narrowing the Gap between Vision and Action in Navigation

## ABSTRACT

The existing methods for Vision and Language Navigation in the Continuous Environment (VLN-CE) commonly incorporate a waypoint predictor to discretize the environment. This simplifies the navigation actions into a view selection task and improves navigation performance significantly compared to direct training using low-level actions. However, the VLN-CE agents are still far from the real robots since there are gaps between their visual perception and executed actions. First, VLN-CE agents that discretize the visual environment are primarily trained with high-level view selection, which causes them to ignore crucial spatial reasoning within the low-level action movements. Second, in these models, the existing waypoint predictors neglect object semantics and their attributes related to passibility, which can be informative in indicating the feasibility of actions. To address these two issues, we introduce a low-level action decoder jointly trained with high-level action prediction, enabling the current VLN agent to learn and ground the selected visual view to the low-level controls. Moreover, we enhance the current waypoint predictor by utilizing visual representations containing rich semantic information and explicitly masking obstacles based on humans' prior knowledge about the feasibility of actions. Empirically, our agent can improve navigation performance metrics compared to the strong baselines on both high-level and low-level actions.

## CCS CONCEPTS

• **Computing methodologies** → *Intelligent agents*.

## KEYWORDS

Vision and Language Navigation (VLN-CE) in the Continuous Environment, Vision and Language, Embodied Agent.

## 1 INTRODUCTION

Vision and Language Navigation (VLN) is a problem setting that requires the agent to navigate in a photo-realistic environment following natural language instructions. Most VLN-related works model the navigation action space as **high-level** viewpoint selection by discretizing the visual environment into images based on an underlying navigability graph, named VLN-DE [5, 11]. Subsequently, the navigation task is extended to the continuous environment (VLN-CE) [23] where the agent navigates with **low-level** controls (*LEFT/RIGHT/FORWARD/STOP*). The VLN-CE agents are

**Unpublished working draft. Not for distribution.**

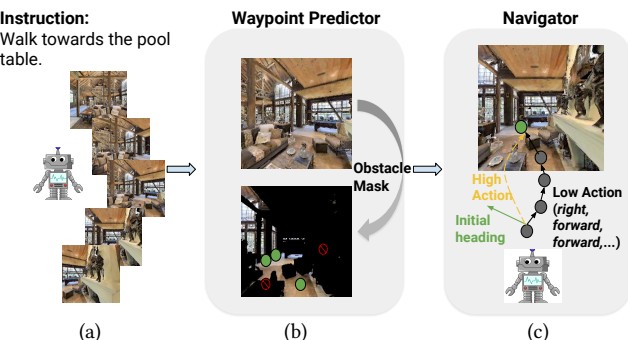

(a)          (b)          (c)

**Figure 1: (a) In the VLN-CE task, instruction and panoramic views of the current navigation step are provided to the agent. (b) We have explored the waypoint predictor considering object semantic and their attributes related to possibility. The green circle is the navigable viewpoints, and the red circle is the obstacle. (c) We equip the navigator with the dual-action module containing both high-level and low-level actions. The black circles are the low-level action sequence.**

closer to the challenges encountered by real-world robots, although their performance is inferior to the VLN-DE agents.

To improve the navigation performance, existing VLN-CE agents employ an offline-trained waypoint predictor, which takes panoramic RGB and depth images as input and generates a local navigability graph centered around the agent at each navigation step [4, 15, 20]. The generated graphs have similar functionality to the connectivity graph in the VLN-DE setting, providing navigable viewpoints. In this case, VLN-CE agents can also use high-level actions (viewpoint selection) to enhance navigation performance. Nevertheless, despite the waypoint predictor's crucial role in existing VLN-CE agents, we have observed two limitations that hinder the navigation agent's understanding of the spatial environment. **First**, the current VLN-CE navigation agents, equipped with the waypoint predictor, are primarily trained to focus on high-level viewpoint selection and rely on an offline controller for low-level action movements within the environment. However, this approach overlooks including low-level actions as part of the training signal. Consequently, the navigation agent misses spatial information embedded in low-level actions, thereby affecting the grounding of different modalities of textual instructions, visual images, and physical spatial motions. **Second**, existing waypoint predictors mainly focus on visual information such as RGB and depth images, overlooking a thorough exploration of object semantic attributes, which are important for assessing the feasibility of the physical actions, such as recognizing that *walls are impassable*.

To address the two above-mentioned issues in the VLN-CE agents with a waypoint predictor, we first introduce a dual-action module in which the agent selects high-level viewpoints while generating low-level action sequences simultaneously. The high-level actions serve as guidance, facilitating the agent's understanding of the relationships between low-level actions and navigable areas indicated

by high-level actions. This enhances the agent's spatial grounding ability to connect action with visual perception and language understanding. Second, to address the issue of the waypoint predictor neglecting object semantic information, we incorporate visual representations with rich object semantics and explicit obstacle masking based on prior knowledge of object passability. As a result, the waypoint predictor will generate more effective viewpoints by leveraging comprehensive pre-training and prior knowledge.

Specifically, we incorporate a Transformer-based decoder into the VLN-CE agent to generate low-level action sequences at each navigation step. We formulate the low-level action generation as a text sequence generation task. As shown in Fig. 1 (c), the agent is flexible in selecting a navigable viewpoint (green circle) from the visual environment or navigating through a low-level action sequence (path with black circle). The low-level action decoder and high-level action selection are jointly trained. This approach encourages the agent to ground the selected view to action planning, bridging the learning gap between low-level and high-level actions, eventually benefiting the navigation performance on both levels. Moreover, to improve the waypoint predictor, we use visual representations obtained from Vision-Language Models Pretrained (VLPMs) [32, 33, 43] containing comprehensive object semantics from different modalities. Besides, we mask obstacle areas in images (such as *"sofa", "table", and "fireplace".* in Fig. 1 (b)) based on the image semantic segmentation and our pre-defined open-area object vocabulary to further encourage the waypoint predictor to generate navigable viewpoints from open areas.

In summary, our contributions are as follows.

1. We introduce a dual-action module for the VLN-CE agents, grounding high-level visual perception into low-level spatial actions. This design empowers the agent with the flexibility to select high-level viewpoints and generate low-level action sequences.

2. We enhance the waypoint predictor with visual representations containing rich object semantics and explicit prior knowledge about objects' passability attributes.

3. We adapt our method on several VLN-CE agents. The experimental results demonstrate the effectiveness of our approach in both waypoint predictor, high-level as well as low-level navigation performance.

## 2 RELATED WORKS

### 2.1 Vision and Language Navigation

Following instructions to navigate in a simulated environment has attracted significant research interest in recent years. There are many navigation datasets introduced, such as indoor navigation R2R [5], R4R [19], RXR [24], outdoor navigation Touchdown [7], object-finding REVERIE [30], and dialogue-based navigation CVDN [40].

Early methods formulate the navigation agent using a sequence-to-sequence framework and apply attention mechanism to improve learning across text and vision modalities [5, 27]. Data augmentation methods are proposed to strengthen the agent generalization ability, such as image editing [25, 26] or instruction generation [11, 49]. The Transformer-based models are the recent main trend for the VLN agent focusing on multi-modal representation

learning [16, 44]. In this framework, map representation learning [2, 8, 17], graph-based exploration [10, 42, 50], and auxiliary pre-training [9, 13, 31, 48, 51] techniques are introduced. However, these agents formulate the navigation in a discrete environment, ignoring the low-level control challenges a real-world robot faces. Our work mainly focuses on improving the agent's performance in the continuous setting.

### 2.2 VLN in Continuous Environment

Several continuous environments have been proposed in the realm of embodied AI to simulate photo-realistic scenes, such as Gibson [47], House3D [46], and Habitat [36]. R2R-CE [22] dataset is constructed by transferring discrete trajectories in R2R [5] dataset to continuous trajectories using Habitat simulator [36]. Compared to discrete environment navigation, VLN-CE is much closer to real-world navigation since it relies on a limited field of view to infer low-level controls rather than a pre-defined connectivity graph with nodes distributed across accessible spaces within a discrete environment.

The VLN-CE is pioneered by [22], and they introduce a seq2seq baseline model, mapping a set of images during navigation to a sequence of low-level actions. LAW [34] improves the baseline using language-aligned waypoint supervisions. The methods applying semantic map [8, 12, 17] are also explored. For example, WS-MGMap [8] proposes a multi-granularity map and introduces a weakly-supervised auxiliary task to learn a better map representation that reveals object information. However, these methods still demonstrate a huge performance gap compared to the navigation agents in the discrete environments. To bridge the gap between VLN-DE and VLN-CE, waypoint models are proposed for hierarchical visual navigation [4, 20]. Specifically, CWP [15] uses a waypoint predictor to provide accessible directions for the agent to act, and then they apply the efficient models in the VLN-DE to the continuous environments. The recent work significantly improves the waypoint models using better visual representations pre-trained in a larger dataset [43, 44], graph-based modeling [1, 2] or extra auxiliary tasks [17]. However, there are still gaps between the visual perception and physical action within the current VLN-CE agent. For example, with the waypoint predictor, the navigation agent is primarily trained based on view selection. The navigation performance drops significantly when modeling low-level actions [15]. In addition, the waypoint predictor mainly focuses on visual features, neglecting rich object semantics within images. Our study focuses on enhancing the VLN-CE agent by addressing these two aspects to narrow the gap by grounding visual perception into spatial actions.

## 3 BACKGROUNDS

### 3.1 Problem Statement

In the VLN-CE problem setting, the navigation agent is required to reach the target location within indoor environments following natural language instructions. Given an instruction, the agent navigates on a 3D mesh of an environment with low-level actions, including RIGHT (15°), LEFT (15°), FORWARD (0.25m), STOP. At each navigation step, the agent observes 12 RGB images and the corresponding depth images from a horizontal panoramic view with 30° intervals. The agent terminates navigation when it selects the *STOP*

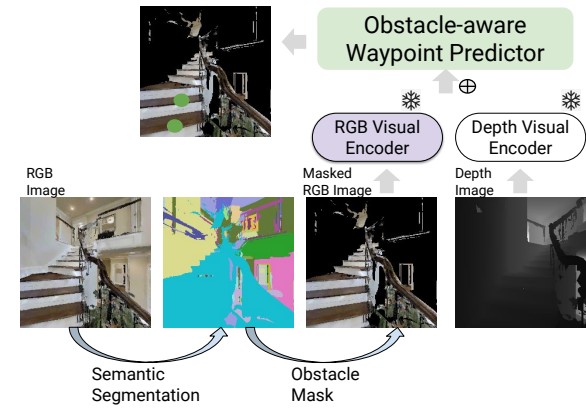

**Figure 2: Obstacle-Aware Waypoint Predictor. Given an RGB image, we mask obstacle objects based on semantic segmentation. The masked RGB image and depth image are then input to the waypoint predictor to generate navigable viewpoints. We also enhance the RGB visual encoder with pre-trained VL representations.**

action or a pre-defined maximum number of navigation steps have been reached.

## 3.2 Backbone Architecture

Our method is designed to be model-agnostic, and we experiment with different VLN-CE agents. In this section, we utilize History Aware Multimodal Transformer (HAMT) in Continuous Environment [9, 43] as the backbone architecture to introduce our method. In the following sections, we first introduce the text and vision encoders. We then present two primary components in the backbone: the *waypoint predictor* and the *navigator*. The waypoint predictor is trained offline to generate navigable viewpoints, which are applied to the navigator for view selection.

**Vision Encoder.** At each navigation step, the agent observes 12 RGB images and 12 depth images, which are represented as $I^{rgb} = \{I_1^{rgb}, I_2^{rgb}, \cdots, I_{12}^{rgb}\}$ and $I^d = \{I_1^d, I_2^d, \cdots, I_{12}^d\}$, respectively. In the baseline, different RGB vision encoders are used for the waypoint predictor and navigator. The waypoint predictor utilizes ResNet-152 [14] as its vision encoder, pre-trained on the ImageNet dataset [35], Meanwhile, the navigator uses the InternVideo [43] as the vision encoder, pre-trained on large video-text datasets. Formally, the obtained visual representations of RGB images are denoted as $v^{rgb} = \{v_1^{rgb}, v_2^{rgb}, \cdots, v_{12}^{rgb}\}$. The depth images are fed into DD-PPO ResNet-50 [45], which is trained for point-goal navigation, represented as $v^d = \{v_1^d, v_2^d, \cdots, v_{12}^d\}$.

**Text Encoder.** The textual instructions are input to the navigator to guide the agent to complete the task. The instruction is denoted as $w = \{w_1, w_2, \cdots, w_l\}$, where $l$ is the length of the instruction tokens. We use BERT [41] to obtain initial text representation of instruction $w$, denoted as $X = [x_1, x_2, \cdots, x_l]$.

**Waypoint Predictor.** We follow the waypoint predictor designed in [15], which is a multi-layer Transformer with a non-linear classifier. All 12 RGB image representations $v^{rgb}$ and depth image representations $v^d$ are concatenated and then input into the waypoint predictor to predict a heatmap of 120 angles-by-12 distances.

Each angle is 3 degrees, and distances range from 0.25 to 3.00 meters with an interval of 0.25 meters. The heatmap is represented as a Gaussian distribution with a variance of 1.75m and 15° to expand the prediction range. The waypoint predictor is pre-trained based on the navigable connectivity graph from MP3D [6]. During inference, non-maximum suppression (NMS) is used to sample $K$ neighboring waypoints, which are utilized as the candidate views for the following navigator.

**Navigator.** HAMT is a multimodal Transformer-based navigator that can memorize historical information. HAMT is initially proposed as a VLN-DE agent. With the introduction of the waypoint predictor, its application is extended to the continuous environment [43, 44]. Specifically, HAMT takes textual instruction, panoramic visual images from the current step, and a sequence of historical panoramic images along the navigation trajectory as input. The output is a selected navigable viewpoint. Formally, at each navigation step $t$, the waypoint predictor generates $K$ candidate views, and their observation representations are denoted as $O_t$. The history representation is denoted as $H_t$. HAMT concatenates history and observation as the vision modality and uses a cross-modal transformer to learn the connection between text presentation $X$ and visual representation $[H_t; O_t]$. The model is trained using high-level viewpoint selection by selecting the highest similarity score between observation encoding $O_t$ and <CLS> token of the Transformer, which contains global instruction-trajectory information. After the agent selects a viewpoint, an offline controller is applied to guide the agent to the corresponding position.

## 4 PROPOSED APPROACH

Built upon the above-introduced backbone architecture, we improve the current VLN-CE agent by introducing an obstacle-aware waypoint predictor and dual-action module for the navigator. Fig. 3 shows the main architecture.

## 4.1 Obstacle-Aware Waypoint Predictor

Object semantics in the visual environment is expected to play an important role in predicting navigable viewpoints, especially in the sense of open and obstacle areas. Objects have attributes that determine whether they should be labeled as passable or impassable. For example, the agent is not supposed to traverse beneath a *"table"* or on the *"bed"*. However, the current methods mainly leverage visual information from RGB and depth images and neglect further exploration of the object semantics and their attributes related to passibility.

To overcome this limitation, we first enhance the current waypoint predictor with vision representation from Vision and Language Pre-trained Models (VLPMs) [32, 33, 43], which contain much more comprehensive object semantics than ResNet used in the baseline's waypoint predictor. We employ vision representations from different VLPMs to assess their influence on the waypoint predictor's performance. Our results demonstrate that CLIP [32] visual representation can achieve the best result. Please see Table 3 for our detailed analysis. Second, we introduce an obstacle mask mechanism based on semantic segmentation within the visual environment and our prior knowledge about impassable objects. We

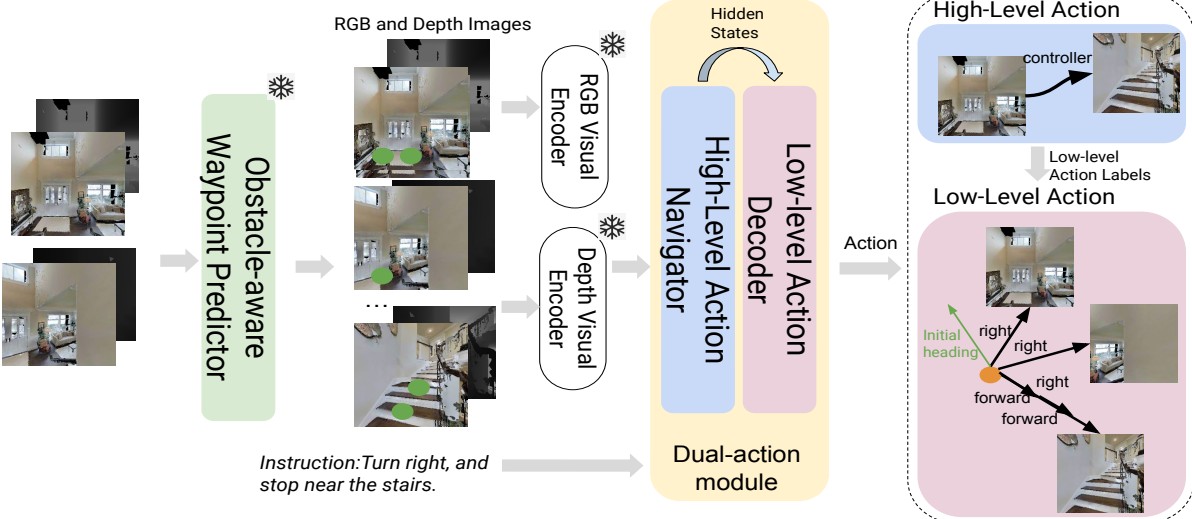

**Figure 3: Main Architecture.** The waypoint predictor first provides navigable viewpoints (green circle). Then, the corresponding RGB images, depth images, and textual instructions are input to our dual-action module, where the agent learns to select a high-level viewpoint and generate a low-level action sequence. The freezing sign indicates that the parameters are freezing during the training process. Please refer to Fig. 4 for a detailed architecture of the low-level action decoder.

utilize semantic segmentation provided by MP3D semantic segmentation with approximately 40 object categories. To identify open areas, we define a vocabulary and mask semantic segments that are not within this vocabulary. For example, we select objects such as *"floor"*, *"stairs"*, and *"door"* as open areas.

Formally, we denote the visual representation from VL pretrained models of panoramic images as $v_c^{rgb} = \{v_{c1}^{rgb}, v_{c2}^{rgb}, \cdots, v_{c12}^{rgb}\}$. For each image, we obtain the corresponding obstacle mask based on semantic segmentation. We assign a label of 1 to object areas in the open vocabulary and 0 otherwise. The resulting obstacle masks are represented as $m = \{m_1, m_2, \cdots, m_{12}\}$. Subsequently, we apply obstacle masks to the RGB images and obtain the masked RGB representation, $v_{cm} = v_c * m$, which is then concatenated with depth visual representation $v^d$. This combined representation is input to the waypoint predictor to generate views at each navigation step.

We train the waypoint predictor with enhanced visual representation and obstacle-masked image. Then, we employ it in the navigator for offline usage to generate navigable views. The details of the navigator are explained in the following section.

## 4.2 Dual-Action Prediction for the Navigator

There are two types of actions in the existing VLN-CE navigator to select: high-level and low-level actions. The current techniques typically model the prediction of these two types of actions independently.

**High-level Action** is to select a view based on the similarity between the observation $O_t$ and the hidden states from the cross-modal Transformer, which is represented as follows,

$$p_t^h = \texttt{Softmax}([H_t; O_t] * h_t^{\texttt{cls}}), \qquad (1)$$

where $h^{\texttt{cls}}$ is the <CLS> token representation from navigator at step $t$, and $p_t^h$ is the probability of high-level action. Once the most similar viewpoint is selected, the agent employs an offline controller to navigate to the corresponding position. While high-level actions effectively boost navigation performance, the training mechanism primarily focuses on view selection, neglecting the spatial information in the low-level action sequence. Additionally, it is challenging for a real-world robot to navigate to a precise angle and distance in a realistic environment. Real-world robots typically operate with very limited action sets, such as FORWARD 0.25m.

**Low-level Action** is to generate low-level action directly. The approach is to use a non-linear classifier to predict an action class at each navigation step. Then, a controller is applied to execute action. Formally, the prediction for low-level actions is performed as follows,

$$p_t^l = \texttt{Softmax}(h_t^{\texttt{cls}} W_c), \qquad (2)$$

where $W_c$ projects the <CLS> token representation to four low-level actions, and $p_t^l$ is the probability of low-level actions. While low-level actions are closer to real-world robotic behavior, directly modeling the agent to generate such actions results in a cost-training process. This is because the episodes for low-level actions are around 10 times longer than high-level actions (around 56 steps for low-level action steps, compared to 4 − 6 for high-level episodes). Additionally, while there are methods [15, 17] attempting to train the navigation agent with a low-level action classifier directly, the performance drops substantially compared to the method using an offline controller.

**Dual-Actions.** We enable the VLN-CE agent to navigate simultaneously using high-level and low-level actions. Built upon the existing VLN-CE agent that predicts high-level actions, we introduce a decoder to generate the corresponding low-level action sequence

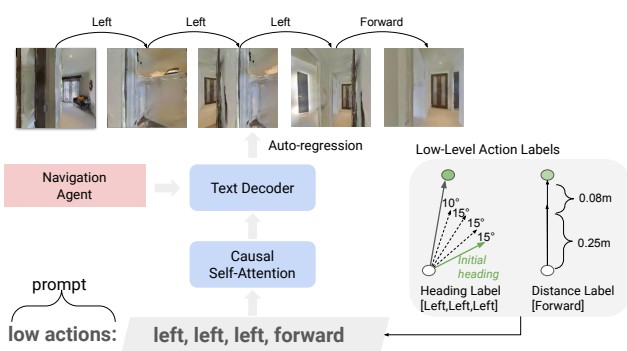

**Figure 4: Low-Level Action Decoder.**

simultaneously. Specifically, instead of generating one low-level action using an action classifier at each navigation step, our agent is trained to generate a low-level action sequence. We formulate low-level action prediction as the textual sequence generation task. As shown in Fig. 4, we introduce a Transformer-based text decoder to generate the low-level action sequences with the prompt of "*low actions:*". At each navigation step, the agent selects a high-level view, and, at the same time, it is trained to generate the corresponding low-level sequence of action tokens auto-repressively. We input the text decoder with <CLS> representation from the navigator and textual prompt representation. The training objective is to maximize the likelihood of the next low-level action token. The following equation shows the loss of generating $j$-th token in the action sequence.

$$\mathcal{L}_{low} = -\sum_j \log p_\theta(a_j^l | a_1^l, \cdots, a_m^i), \qquad (3)$$

where $m$ is the length of the sequence.

We jointly train low-level action decoders and high-level selection. The labels for the low-level action sequence are obtained from the heading and distance differences between the initial view and the selected views in the high-level action. When the agent selects a high-level view at each navigation step, the corresponding low-level sequence label is created. Specifically, we initially calculate the degree difference in headings and divide it by 15 to determine the number of rotation steps the agent takes. The direction (LEFT or RIGHT) is determined based on the smaller rotations. A similar process is applied to distance: we calculate the distance between the start point and the selected view and divide it by 0.25. We ignore the remaining if the heading or distance is not perfectly divisible. For instance, as shown in Fig. 4, the distance between the start point and the target view is 0.33m, and the low action is just one FORWARD (0.25m).

## 5 TRAINING AND INFERENCE FOR NAVIGATOR

For the high-level action, we train the navigator with a mixture of Imitation Learning (IL) and Reinforcement Learning (RL) [39]. It consists of the cross-entropy loss of the predicted probability distribution against the ground-truth action and a sampled action from the predicted distribution to learn the designed rewards. In summary, the navigation loss is as follows,

$$\mathcal{L}_{high} = -\sum_t -\alpha_t^h log(p_t^h) - \lambda \sum_t \alpha_t^s log(p_t^h), \qquad (4)$$

where $\lambda$ is the hyper-parameter to balance the two components, $\alpha_t^h$ is the teacher action for IL that is closest to the next viewpoint, and $\alpha_t^s$ is a sample action for RL. The joint training objective for our high-level action navigator with the low-level action decoder is as follows,

$$\mathcal{L} = \mathcal{L}_{high} + L_{low}. \qquad (5)$$

During inference, in terms of navigating with high-level action, we use greedy search to select an action with the highest probability at each navigation step to generate a trajectory. For navigation with low-level action, we provide the text prompts "*low actions:*" to the low-level action decoder and utilize a beam search approach to generate the low action sequence. Then, we apply post-processing to separate all actions and use a controller to execute each action. For the next navigation step, the agent relies on the previous low-level action results, even though it is trained based on view selection.

## 6 EXPERIMENTS

### 6.1 Dataset and Evaluation Metrics

VLN-CE uses the Habitat Simulator [29, 37, 38] to render environment observations based on the MP3D dataset [6], including 61 environments for training, 11 for unseen validation and 18 for testing. The scenes in the validation unseen and test unseen sets differ from those in the training set. Three main metrics are used to evaluate navigation performance. 1) Success Rate (SR) [5] is to evaluate wayfinding performance to indicate whether the agent arrives at the destination. 2) Success Rate Weighted Path Length (SPL) [3] normalizes success rate by trajectory length. 3) normalized Dynamic Time Warping (nDTW) [18] is used to measure the fidelity between the predicted and the ground-truth trajectories.

### 6.2 Implementation Details

The waypoint predictor is trained on a single NVIDA RTX GPU for 20 hours, while the navigator is trained on 6 NVIDA RTX GPUs for 2.5 days. We adopt the model architecture and implementation from [9, 43] based on the PyTorch framework [28] and the Habitat Simulator [36]. The RGB image size is 224 and the depth image size is 256. CLIP ViT-B/16 is used as visual input for the waypoint predictor. We train the waypoint predictor with 300 epochs with a batch size of 8 and a learning rate of $1e-6$. We train the navigator 10000 iterations with the batch size of 6 and the learning rate of $3e-5$. The $\lambda$ in Eq. 4 is 0.75. For the low-level action generator, we set the maximum generated length as 30 and the beam search size as 3. The end token is the comma.

### 6.3 Experimental Results

We evaluate our method on top of three VLN-CE agents: WP-VLN-BERT [15], WP-HAMT [43], and ETPNav [2]. These VLN-CE agents are all Transformer-based models and employ a waypoint predictor to discretize the visual environment for navigation in the continuous setting. WP-VLN-BERT uses an implicit state representation to store historical information, whereas WP-HAMT improves this WP-VLN-BERT by storing historical panoramic images from the

| | Model | Validation Unseen | | | Test Unseen | |
|---|---|---|---|---|---|---|
| | | nDTW↑ | SR↑ | SPL↑ | SR↑ | SPL↑ |
| 1 | Waypoint Models [20] | - | 0.36 | 0.34 | 0.32 | 0.30 |
| 2 | CWP-CMA [15] | 0.55 | 0.41 | 0.36 | 0.38 | 0.33 |
| 3 | Sim2Sim [21] | - | 0.43 | 0.36 | 0.44 | 0.37 |
| 4 | VLN-BERT+Ego2-Map [17] | 0.60 | 0.52 | 0.46 | 0.47 | 0.41 |
| 5 | WP-VLN-BERT [15] | 0.54 | 0.44 | 0.39 | 0.42 | 0.36 |
| 6 | **WP-VLN-BERT+Ours** | 0.55 | 0.46 | 0.41 | 0.44 | 0.38 |
| 7 | WP-HAMT [43] | 0.60 | 0.52 | 0.47 | 0.49 | 0.45 |
| 8 | **WP-HAMT+Ours** | **0.62** | 0.54 | **0.49** | 0.52 | 0.47 |
| 9 | ETPNav [2] | - | 0.57 | **0.49** | 0.55 | 0.48 |
| 10 | **ETPNav+Ours** | **0.62** | **0.58** | **0.49** | **0.56** | **0.48** |

**Table 1: Experimental results on high-level action evaluated on the R2R-CE validation unseen and test dataset.**

traversed path. ETPNav is a graph-based VLN agent, which differs slightly from the standard navigation setting. The other two agents can only select local navigable viewpoints connected to the current viewpoint. However, graph-based agents like ETPNav can jump back to the previously explored viewpoints, often resulting in a higher success rate. Please refer to the supplementary for a more detailed introduction for each baseline. We integrate our dual-action module and enhanced waypoint predictor into the three baseline backbones introduced above and evaluate navigation performance on both high-level and low-level actions as follows.

**High-Level Action Performance.** Table 1 presents the navigation performance using high-level actions on the validation unseen and test unseen sets. All VLN-CE agents in Table 1 employ a waypoint predictor to generate navigable views and are trained with the high-level view selection. Our method improves high-level action performance for all baseline agents. Specifically, we can improve WP-VLN-BERT almost 2% on all navigation metrics on both validation and test unseen sets, as shown in row#6. WP-HAMT utilizes visual representation from InternVideo [43] to strengthen the model's performance. We mainly compare with InternVideo base weights because of the computation cost limitation. In terms of this baseline, we can improve the navigation performance of the baseline, especially 3% of the success rate on the test unseen (row#8). In addition to enhancing the standard Transformer-based navigation agent, our method can also increase the success rate of the graph-based agent ETPNav, as shown in row#10.

**Low-Level Action performance.** Table 2 shows the navigation performance with low-level movement actions. The existing methods mainly compare the results of the validation unseen for low-level actions. The models in row#1 to row#3 are based on LSTM architecture to frame the navigation as a sequence-to-sequence task and predict low-level actions directly. The models from row#4 to row#6 are models using a waypoint predictor. They add an action classifier to the navigator and train the model to select one low-level action at each navigation step. Another noticeable difference between the models from row#1 to row#3 and others is that the agent in these methods can only observe the current view rather than the whole panoramic view at each navigation step. As shown in Table 2, we observe that some Transformer-based navigators with much more powerful pre-trained visual representations and complex model architecture, such as the approaches in row#5 and row#6, their low-level action navigation performance could not compete with LSTM-based models (row#1 to row#3). Specifically,

| | Methods | nDTW↑ | SR ↑ | SPL ↑ |
|---|---|---|---|---|
| 1 | CMA+PM+DA+Aug [23] | 0.51 | 0.32 | 0.30 |
| 2 | LAW [34] | 0.54 | 0.35 | 0.31 |
| 3 | WS-MGMap [8] | - | 0.39 | 0.34 |
| 4 | CWP-CMA [15] | 0.49 | 0.27 | 0.25 |
| 5 | VLN-BERT+Ego2-Map [17] | 0.52 | 0.30 | 0.29 |
| 6 | WP-VLN-BERT [15] | 0.48 | 0.23 | 0.22 |
| 7 | **WP-VLN-BERT+Ours** | 0.54 | 0.28 | 0.27 |
| 7 | WP-HAMT* [43] | 0.54 | 0.35 | 0.32 |
| 9 | **WP-HAMT+Ours** | 0.55 | 0.44 | 0.38 |
| 10 | **ETPNav+Ours** | **0.58** | **0.48** | **0.42** |

**Table 2: Experimental results of low-level actions on the R2R-CE validation unseen set. * means our implementation for low-level action prediction, as most VLN-CE agents do not report their low-level performance. We train the VLN agent with a low-level action classifier for fair comparison.**

| | Visual Encoder | Waypoint Predictor | | | |
|---|---|---|---|---|---|
| | | $|\Delta|$ | %Open↑ | $d_C \downarrow$ | $d_H \downarrow$ |
| 1 | ResNet [15] | 1.40 | 0.80 | 1.07 | 2.00 |
| 2 | InternVideo [43] | 1.44 | 0.65 | 1.15 | 2.04 |
| 3 | DenseCLIP [33] | 1.41 | 0.81 | 1.05 | 2.01 |
| 4 | CLIP [32] | **1.38** | 0.83 | **1.04** | 2.00 |
| 5 | CLIP+ Obstacle Mask | **1.38** | **0.85** | **1.04** | **1.94** |

**Table 3: Evaluation of different VLM RGB visual encoders.**

for the baseline model WP-VLN-BERT, although our method can significantly improve it (row#7), it is still far behind LSTM-based models. However, we can achieve SOTA after applying our method on WP-HAMT and ETPNav, as shown in row#9 and row#10, respectively. It is worth noting that the majority of VLN-CE agents do not report their low-level action performance. To address this, we follow the method of low-level action prediction in WP-VLN-BERT to add a non-linear classifier on top of WP-HAMT [43] to adapt it to low-level action prediction. In general, our method's performance is aligned with the VLN-CE navigator's performance. This correlation is expected, as the low-level action sequence is trained using the hidden state representation from the corresponding baseline, and stronger representations yield better performance when training low-level actions.

## 6.4 Ablation Study

In this section, we conduct an ablation analysis of the waypoint predictor and for waypoint predictor and the effectiveness of different components in our method.

**Waypoint Predictor Performance.** The waypoint predictor is trained offline to generate navigable viewpoints. We enhance the baseline waypoint predictor from the aspects of stronger visual representations and explicit object masks, and Table 3 shows our results on R2R-CE validation unseen set. The main metrics to evaluate the waypoint predictor's performance are as follows: $|\Delta|$ measures the difference in the number of target waypoints and predicted waypoints. %Open is the ratio of predicted waypoints in open space. $d_c$ and $d_H$ are the Chamfer and Hausdorff distances, respectively, to measure the distance between point clouds.

| Method | | | | High | | | Low | | |
|---|---|---|---|---|---|---|---|---|---|
| | CLIP | Ob-Mask | Dual-Action | nDTW↑ | SR↑ | SPL↑ | nDTW↑ | SR↑ | SPL↑ |
| Baseline | | | | 0.60 | 0.52 | 0.47 | 0.54 | 0.35 | 0.32 |
| 1 | | | ✔ | 0.60 | 0.52 | 0.47 | **0.55** | 0.43 | 0.36 |
| 2 | ✔ | | | 0.61 | 0.53 | 0.47 | - | - | - |
| 3 | ✔ | ✔ | | 0.61 | 0.53 | 0.48 | - | - | - |
| 4 | ✔ | ✔ | ✔ | **0.62** | **0.54** | **0.49** | **0.55** | **0.44** | **0.38** |

**Table 4: Ablation study on different components of our method. The baseline is WP-HAMT, and the Ob-mask is the obstacle mask.**

| | Ob-Mask Vocab | **Waypoint Predictor** | | | |
|---|---|---|---|---|---|
| | | $|\Delta|$ | %Open↑ | $d_C \downarrow$ | $d_H \downarrow$ |
| | No Mask | **1.38** | 0.83 | **1.04** | 2.00 |
| 1 | Floor | **1.38** | 0.84 | 1.07 | 2.00 |
| 2 | Stairs | 1.40 | 0.82 | 1.04 | 2.00 |
| 3 | Doors | 1.40 | 0.80 | 1.04 | 1.94 |
| 4 | Floor+Stairs+Doors | **1.38** | **0.85** | **1.04** | **1.94** |

**Table 5: Analysis of the influence of various open-area vocabularies on the waypoint predictor.**

We experiment with visual representations from different Vision and Language Pre-trained Models (VLMs) and test their influence on the performance of the waypoint predictor. As shown in Table 3, the waypoint predictor achieves the best performance when utilizing CLIP vision representations. However, we cannot conclude that more powerful vision representations lead to better waypoint-predicting performance since the representations from InterVideo seem to hurt the waypoint predictor. This result suggests that different pre-trained visual encoders possess varying capacities to influence the agent's ability to recognize open and obstacle areas. The better result is achieved when both CLIP representation and our designed obstacle mask are applied. Compared to ResNet (row#1), CLIP improves the open area prediction by about 3%, demonstrating that rich semantics in visual representation in CLIP aids the waypoint predictor in learning the open and obstacle objects. After applying our designed obstacle mask, the accuracy of open area prediction gained an additional improvement of 2%, emphasizing the effectiveness of the prior knowledge in encouraging the waypoint predictor to better focus on open area spaces.

**Different Components.** The results of the ablation study in Table 4 demonstrate the influence of each component of our proposed method on both high-level and low-level navigation. The components in our method include dual-action for the navigator and the enhanced waypoint predictor with CLIP visual representations and the obstacle mask. The analyzed navigator is WP-HAMT, which uses visual representation obtained from InternVideo base weights. We report results on the R2R-CE validation unseen dataset. In row#1, we integrate the low-level action decoder with the baseline navigator and jointly train it with high-level actions, and the low-level action navigation performance is significantly improved (about 4% on SPL). In row#2, we train the waypoint predictor with only CLIP representations and apply it to the baseline navigator without dual-action training. Notably, the enhanced waypoint predictor already contributes to better high-level navigation performance, indicating that CLIP's rich object semantic representation boosts overall navigation. Row#3 shows the effectiveness of the obstacle mask in enhancing the SPL for high-level action. In row#4, we train

(1) Instruction: Turn around stairs, and walk towards the living room.

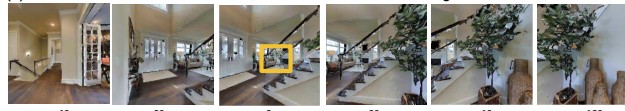

| -60 | -30 | 0 | 30 | 60 | 120 |

**Low Action**: *forward, forward, forward, forward, forward.*

(2) Instruction: Turn around the table and turn left to the kitchen.

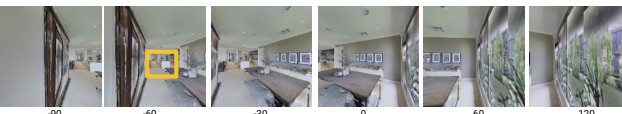

| -90 | -60 | -30 | 0 | 60 | 120 |

**Low Action**: *left, left, left, left, forward, forward, forward, forward, forward, forward, forward, forward.*

**Figure 5: Examples of generated low-level actions.** 0 **denotes the current direction, while** − **means** LEFT **turn. The number represents the rotation degree. The yellow bounding box indicates the target.**

the waypoint predictor with both CLIP and obstacle mask, and we apply this enhanced predictor to the navigator with the dual-action module. We achieve the best results in this setting. Compared to row#3, we conclude that the spatial information incorporated into the low-level action benefits the high-level viewpoint selection. Similarly, compared to row#1, enhanced waypoint prediction not only enhances high-level viewpoint selection but also benefits low-level action generation.

## 6.5 Qualitative Analysis

In this section, we provide a qualitative analysis from the perspectives of low-level actions and obstacle masks.

**Low-Level Actions Generation.** In Fig. 5 (a), we show an example of our generated low-level action sequences that lead the agent to the destination. However, we have observed cases where the agent generates a low-level action sequence that reaches the destination but does not fully follow the instruction, as shown in Fig. 5 (b). This issue also occurs in other VLN-CE agents when modeling low-level action predictions. We assume the reason is the inherent challenges in building the VLN-CE dataset. It transfers the instructions and trajectories from VLN-DE. When the simulator executes the low-level actions, especially rotations, there is no human evaluation process to confirm whether the low-level actions align with the instruction. The actions are generated based on minimal required rotations when the selected view is given to the simulator. Training the agent to learn low-level actions in these scenarios is challenging compared to directly training with view selection.

**Open-area Vocabulary.** In Table 5, we provide an analysis of object semantics and their relation with the performance of the waypoint predictor. We select open-area vocabularies based on our prior knowledge. The baseline we used is the waypoint predictor trained with CLIP visual representations. Given the semantic segmentation from the simulated environment, we mask other object areas except the semantic areas in open-area vocabularies. Then, we input the masked image into the waypoint predictor. The experimental results demonstrate that different object semantics show varying influences on the waypoint predictor. For instance, the %open is low when we mask objects other than "*door*", indicating the presence of closed or blocked doors (row#3). We can get the best

## (1) RGB Image

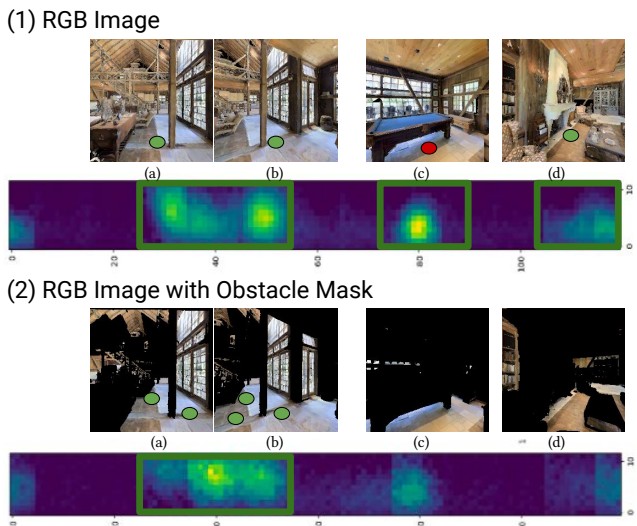

## (2) RGB Image with Obstacle Mask

**Figure 6: An example of a generated waypoint heatmap given an RGB image with and without obstacle mask. The x and y axes represent heading and distance. The green box highlights the areas where the model samples waypoints. The green and red circles indicate waypoints within or outside the ground-truth target areas.**

results when our open-area vocabularies contain "*floor*", "*stairs*", and "*doors*" (row#4).

**Qualitative Examples for Obstacle Mask.** In Fig. 6, we show an example to demonstrate the different generated waypoint heatmaps between an RGB image with obstacle mask (Fig. 6(2)) and without it (Fig. 6(1)). The image shows the corresponding views based on the headings of the highlighted areas in the heatmap. It is evident that the waypoint predictor samples more viewpoints (5 viewpoints) from image (a) and image (b) when an obstacle mask is applied, both of which contain large open areas. In contrast, RGB images without obstacle masks sample relatively fewer viewpoints on images (a) and (b), but they sample viewpoints from (c) and (d), although (c) is not included in the ground truth. This example illustrates that the obstacle mask aids the waypoint predictor in concentrating mainly on large open areas but falls short in narrow open areas. However, based on the final navigation result in Table 4, the obstacle mask ultimately contributes to navigation performance.

## 7 CONCLUSION

In this work, we narrow the gap between the vision and action of the current VLN-CE agents mainly in two aspects. we introduce a dual-action module that enables the current VLN-CE agent, equipped with the waypoint predictor, to jointly train for both high-level and low-level actions simultaneously. This joint optimization encourages the agent to learn to ground the high-level visual perception and view selection into physical actions and spatial motions. Second, we enhance the existing waypoint predictor by incorporating rich object semantic representations and knowledge about object attributes. This helps the model to understand the feasibility of actions better. We conduct comprehensive experiments and analyses to demonstrate the effectiveness of our proposed method.

## LIMITATIONS

We mainly summarize the following limitations. First, integrating the low-level action decoder into the current VLN-CE agent leads to increased computational costs compared to the baseline. However, the significant improvement in low-level action performance partially justifies this tradeoff. Second, although our designed dual-action module improves the low-level action navigation, the performance gap still remains compared to high-level action performance.

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
