# OpenReview forum: "Narrowing the Gap between Vision and Action in Navigation"
_acmmm.org/ACMMM/2024/Conference — MM2024 Poster_

### Official Review · Reviewer_7rqw · 2024-05-20

**Rating:** 5
**Confidence:** 4

**Summary:**

This paper proposes two useful modules to narrow the gap between the vision and action of the current VLN-CE agents. The first module is Obstacle-aware Waypoint Predictor and the second module is the Dual-Action module. Based on the ablation study, both modules are effective.

**Strengths:**

- The design of the Low-Level Action Decoder is creative. This module connects high-level action and low-level action, which narrows the gap between VLN-DE and VLN-CE.
- The experiment results are persuasive.

**Limitations:**

- The reviewer would like to confirm if the high-level performance in Table 1 only is related to the Obstacle-aware Waypoint Predictor.
- Some typos: There is no (a) (b) in Fig 5. The Fig.5 (a) in line 785 and the Fig.5 (b) in line 790 are nonsense.
- The title of the article seems a bit too broad. This work just narrows the gap between high-level action and low-level action, instead of vision and action.

**Suitability:**

3

---

### Official Review · Reviewer_bmZB · 2024-05-24

**Rating:** 2
**Confidence:** 4

**Summary:**

They think the VLN-CE agents are still far from the real robots since there are gaps between their visual perception and executed actions. To address the problem in the VLN-CE agents with a waypoint predictor, They introduce a dual-action module in which the agent selects high-level viewpoints while generating low-level action sequences simultaneously.  They enhance the waypoint predictor with visual representations containing rich object semantics and explicit prior knowledge about objects’ passability attributes.

**Strengths:**

1. They introduce a low-level action decoder, enabling the current VLN agent to learn and ground the selected visual view to the low-level controls.

2. They enhance the current waypoint predictor by utilizing visual representations containing rich semantic information and explicitly masking obstacles based on humans’ prior knowledge about the feasibility of actions.

**Limitations:**

1. They believe the VLN-CE agents are still far from the real robots since there are gaps between their visual perception and executed actions. But now there are also some VLN datasets that use low-level actions, such as Touchdown, map2seq. I don't understand the significance of this paper.

**Suitability:**

3

---

### Official Review · Reviewer_77dR · 2024-05-25

**Rating:** 4
**Confidence:** 3

**Summary:**

This paper focuses on solving the gap between visual perception and executed actions in VLN-CE tasks. By introducing a dual-action module for the VLN-CE agents, grounding high-level visual perception into low-level spatial actions and enhancing the waypoint predictor with visual representations, this paper increases the high-level navigation performance and low-level navigation performance.

**Strengths:**

1: The proposed dual-action module is novel. It views low-level actions as tokens in NLP and generate low-level action sequence from heading and distance differences between views in high-level action.
2: The paper provides an evaluation of the system with different baselines to showcase the effectiveness of the system.

**Limitations:**

Insufficient evaluation:
1: Real-time performance of the system is not evaluated, considering the paper focuses on mitigating the gap between VLN-CE agent and real robots. Analysis on inference efficiency is persuasive.

2: In Section 6.2 "Implementation Details", the author writes "The waypoint predictor is trained on a single NVIDA RTX GPU for 20 hours, while the navigator is trained on 6 NVIDA RTX GPUs for 2.5 days.". It will be more Intuitive if the author can list the GPU model, or at least the computing power.

Some points were not made clear: How the labeling process of the low-level action sequence handling the obstacles between initial view and the selected views in the high-level action? Since the labels only use the heading and distance differences between views, thus the low-level actions cannot avoid obstacles while moving.

**Suitability:**

3

---

### Meta-Review · Area_Chair_riTo · 2024-07-06

**Recommendation:** Accept (Poster)
**Confidence:** 5

**Metareview:**

This paper proposes to tackle the issues of existing vision and language navigation methods in ignoring low-level spatial reasoning and high-level object semantics. To this end, an obstacle-aware waypoint predictor and a dual-action module for selecting high-level viewpoint and low-level action sequences are introduced. These modules are combined with several agents for evaluation and the experimental results show they indeed bring performance gains.

The pre-rebuttal scores of this paper were mixed, including one weak reject, one borderline accept, and one weak accept. The major concern from the reviewers was about the clarification of the proposed method, which was addressed in the rebuttal. The post-rebuttal ratings were increased to two weak accepts and one borderline accept. The AC also agrees with the reviewers’ assessments and recommends accepting the paper.